# The Effect of Light Source Line Width on the Spectrum Resolution of Dual-Frequency Coherent Detection Signals

**DOI:** 10.3390/s19235264

**Published:** 2019-11-29

**Authors:** Jianying Ren, Huayan Sun, Laixian Zhang, Yanzhong Zhao

**Affiliations:** 1Graduate School, Space Engineering University, Beijing 101416, China; 2Department of Electronic and Optical Engineering, Space Engineering University, Beijing 101416, China; shy221528@vip.sina.com (H.S.); zhanglaixian@126.com (L.Z.); yzzhao1984@sina.com (Y.Z.)

**Keywords:** dual-frequency coherence, line width, doppler shift, power spectrum, microwave beat frequency

## Abstract

In this paper, the power spectrum resolution problem of dual-frequency coherent mixing signals is analyzed when the Doppler frequency difference is small. The power spectrum function formula of the four optical coherent mixing signals is obtained using statistical theory and the Wiener–Khinchin theorem. The influence of delay time and light source line width on the power spectrum of dual-frequency coherent signals is analyzed using this formula. The results show that delay time only affects the peak of the power spectrum of the coherent signal. An increase in the line width of the light source broadens the signal power spectrum and reduces the peak value. The necessary condition for distinguishing the Doppler frequency difference is that the theoretical Doppler frequency difference is greater than 1/5 times the line width of the light source.

## 1. Introduction

Compared to direct detection technology, laser coherent detection technology greatly improves the signal-to-noise ratio and detection sensitivity. At the same time, it has the advantage of a strong anti-interference ability, and is widely used in target detection [1], synthetic aperture imaging [2], Doppler measurement [3,4], and other fields. With the development of technology and changes in demand, single-frequency laser coherent detection cannot meet the measurement requirements of higher precision and higher speed.

Dual-frequency laser coherent detection technology can reduce the high Doppler shift of high-speed moving targets to microwave frequency by the microwave beat method [5,6], which solves the contradiction between the low response frequency and larger Doppler frequency shifts. Therefore, dual-frequency laser coherent detection is useful in high-precision measurement and ultra-high-speed measurement, and has become a popular research direction in the field of coherent detection [7,8,9,10]. The important factors affecting the sensitivity, stability and resolution of coherent detection are source intensity fluctuation, phase noise and line width. The main method to reduce these effects is to use a narrow line width, high stability mode-locked laser [11,12,13].

The Doppler shift resolution of dual-frequency light is key to dual-frequency laser coherent detection. In the case of high-speed motion and a large amount of frequency shift, the Doppler frequency difference of the dual-frequency light is large, and the Doppler shift resolution of the dual-frequency light is easy to implement. However, in the application process, in cases where the motion speed or the frequency shift is small, the Doppler frequency difference will be small, which makes the Doppler spectrum difficult to distinguish. Although this is a special situation, it is necessary to understand the correlation between Doppler shift resolution, Doppler frequency difference, and source line width. However, there is still a lack of theoretical analysis regarding this aspect.

In this paper, we use statistical theory and the Wiener–Khinchin theorem to obtain the power spectrum function of dual-frequency light, and analyze the relationship between the Doppler spectrum of dual-frequency light and line width, frequency shift, delay time and motion speed. It provides theoretical support and reference for the research and application of dual-frequency laser coherent detection technology.

## 2. Theoretical Analysis of Four-Light Coherent Mixing Technology

The structure of the dual-frequency laser coherent detection system is shown in Figure 1.The time domain signal of the dual-frequency local oscillator can be expressed as $E_{01}\left(t\right)$ and $E_{02}\left(t\right)$, where:(1)$$\begin{array}{ll} E_{0} & =\left[E_{01}\left(t\right)+E_{02}\left(t\right)\right]\\ & =bE_{0}\cos\left[\omega_{1}t+\varphi_{01}\left(t\right)\right]+\left(1-b\right)\eta E_{0}\cos\left[\omega_{2}t+\varphi_{02}\left(t\right)\right] \end{array}$$
$E_{0}$ is the amplitude of the time domain light field. $\omega_{1}$ and $\omega_{2}$ are the angular frequency, and $\omega_{2}=\omega_{1}+\omega_{IF}$, $\omega_{IF}$ is the frequency shift. It can be assumed that the frequency shift of the acousto-optic frequency shifter (AOFS) is positive. $\varphi_{01}\left(t\right)$ and $\varphi_{02}\left(t\right)$ are the random phase of the light field, and *b* is the amplitude ratio of the light field, $E_{01}\left(t\right)$ and $E_{02}\left(t\right)$. η is the frequency shifter diffraction efficiency.

The time domain signal of the reflected light of the target received by the receiving system can be expressed as:(2)$$\begin{array}{ll} E_{R} & =E_{s1}\left(t+\tau_{d}\right)+E_{s2}\left(t+\tau_{d}\right)\\ & =\beta E_{0}\cos\left[\omega_{1}\left(t+\tau_{d}\right)+\delta_{1}t+\varphi_{01}\left(t+\tau_{d}\right)\right]\\ & \;+\beta E_{0}\cos\left[\omega_{2}\left(t+\tau_{d}\right)+\delta_{2}t+\varphi_{02}\left(t+\tau_{d}\right)\right] \end{array}$$
$E_{s1}$ and $E_{s2}$ are the reflected light field signals. β is the amplitude ratio of the local oscillator signal to the detected signal. $\tau_{d}=2R/c$ is the time delay of the detection signal relative to the local oscillator signal. R is the distance and *c* is the speed of light. $\delta_{1}$ and $\delta_{2}$ are the Doppler shift, which is related to the speed and frequency of motion.
(3)$$\delta_{1}=2V\omega_{1}/c\;,\;\delta_{2}=2V\omega_{2}/c$$
V is the target speed. When the target moves toward the detection system, the velocity (*V*) is positive. When moving away from the detection system, the velocity (*V*) is negative. In this paper, it is assumed that the target is moving forward along the x-axis, meaning the Doppler shift is positive.

The detection source and the local oscillator are coherently mixed on the surface of the detector, and the optical mixing signal is converted into a digital signal by the detector. Theoretically, the dual-frequency detection signal and the dual-frequency reference signal are mixed to produce a sixth-order component, where the amplitude of the coherent term of the dual-frequency echo signal is small and can be ignored. The interference term includes a frequency shift term, $\omega_{IF}$, and a heterodyne coherent term, $\omega_{IF}\pm \delta_{n}$. When the frequency shift $\omega_{IF}$ is greater than the detector cutoff response frequency, the detector cannot respond. Only the mixing component containing the Doppler shift terms $\delta_{1}$ and $\delta_{2}$ is what we hope to obtain.
(4)$$U_{1}\left(t\right)=b\beta E_{0}^{2}\cos\left[\delta_{1}t+\omega_{1}\tau_{d}+\varphi_{01}\left(t+\tau_{d}\right)-\varphi_{01}\left(t\right)\right]$$
(5)$$U_{2}\left(t\right)=\left(1-b\right)\beta \eta E_{0}^{2}\cos\left[\delta_{2}t+\omega_{2}\tau_{d}+\varphi_{02}\left(t+\tau_{d}\right)-\varphi_{02}\left(t\right)\right]$$
Theoretically, the coherent optical field of Equation (4) and the coherent optical field of Equation (5) can be obtained by the second mixing, remove the DC component.
(6)$$U_{mix}\left(t\right)=b\left(1-b\right)\beta^{2}\eta E_{0}^{4}\cos\left[\Delta \delta t+\omega_{IF}\tau_{d}+\Delta \varphi_{01}\left(t\right)-\Delta \varphi_{02}\left(t\right)\right]$$
$\Delta \delta =\left|\delta_{2}-\delta_{1}\right|$ is the beat frequency of the Doppler shift. For ease of analysis, assume $bE_{0}^{2}=\left(1-b\right)\eta E_{0}^{2}=1$.

Resolving the spectrum of two light field signals in the frequency domain is a necessary condition for generating a microwave beat signal. In practical applications, the light source has a certain line width. In addition, due to the influence of laser noise and environmental noise, the spectrum of the coherent signal is broadened, which increases the difficulty of spectrum discrimination. In this paper, the influence of the line width of the light source on the spectral resolution performance of Equations (4) and (5) is analyzed, under the condition of a small Doppler shift.

## 3. Coherent Mixing Signal Power Spectrum Analysis

Consider the coherent mixing signal as a random stationary signal. According to the Wiener–Khinchin theorem, the autocorrelation function of the stochastic stationary signal and the corresponding power spectral density function are Fourier transform pairs [13].

Amplitude noise contribution to the field spectrum can be neglected, in spite of its large spectrum width, because its integration power is much smaller than its phase noise. The autocorrelation function of the light field can be expressed as [13,14].
(7)$$G\left(\tau \right)=\langle U\left(t\right)U^{*}\left(t+\tau \right)\rangle$$

The time average is expressed as 〈 〉. Consider $U_{1}(t)$ and $U_{2}(t)$ as mutually independent coherent 
mixing signals, and the autocorrelation function is [15,16]: (8)$$\begin{array}{l} G\left(\tau \right)=\langle U_{1}\left(t\right)U_{1}^{*}\left(t+\tau \right)\rangle +\langle U_{2}\left(t\right)U_{2}^{*}\left(t+\tau \right)\rangle \\ \;=A^{2}+\frac{1}{2}\beta^{2}\cos\left(\delta_{1}\tau \right)\langle \varphi_{01}\left(t+\tau_{d}+\tau \right)-\varphi_{01}\left(t+\tau_{d}\right)-\varphi_{01}\left(t+\tau \right)+\varphi_{01}\left(t\right)\rangle \\ \;+A^{2}+\frac{1}{2}\beta^{2}\cos\left(\delta_{2}\tau \right)\langle \varphi_{02}\left(t+\tau_{d}+\tau \right)-\varphi_{02}\left(t+\tau_{d}\right)-\varphi_{02}\left(t+\tau \right)+\varphi_{02}\left(t\right)\rangle \end{array}$$
Δφ(t,τ)=φ(t+τ)−φ(t) expresses the correlation of random phases at different times, and can also be understood as the amount of phase change in time (τ). $A=1+\beta^{2}$. According to the theory of signal noise, the random phase change of a signal is a zero-mean Gaussian random process [14,15].
(9)$$\langle \exp\left[\pm i\Delta \varphi \left(t,\tau \right)\right]\rangle =\exp\left[-\langle \Delta \varphi^{2}\left(\tau \right)\rangle /2\right]$$
$\langle \Delta \varphi^{2}(\tau )\rangle$
is the random phase change variance. According to the literature [17,18], the random phase variance is
(10)$$\sigma^{2}=\langle \Delta \varphi^{2}\left(\tau \right)\rangle =\Delta w\left|\tau \right|=2\frac{\left|\tau \right|}{\tau_{c}}$$
Δw is the full half-high width of the characteristic line. $\tau_{c}=1/\pi \Delta f$ is the coherence time of the laser source [13,14].
(11)$$G\left(\tau \right)=\left\{ \begin{array}{l} 2A^{2}+\frac{1}{2}\beta^{2}\cos\left(\delta_{1}\tau \right)\exp\left(-\Delta w\tau_{d}\right)\\ \;+\frac{1}{2}\beta^{2}\cos\left(\delta_{2}\tau \right)\exp\left(-\Delta w\tau_{d}\right)\;\tau \geq \tau_{d}\\ 2A^{2}+\frac{1}{2}\beta^{2}\cos\left(\delta_{1}\tau \right)\exp\left(-\Delta w\left|\tau \right|\right)\\ \;+\frac{1}{2}\beta^{2}\cos\left(\delta_{2}\tau \right)\exp\left(-\Delta w\left|\tau \right|\right)\;\tau <\tau_{d} \end{array}\right.$$
The autocorrelation function of the random signal is an even function. According to the Wiener–Khinchin theorem [13], the power spectrum of the autocorrelation function is
(12)$$G\left(\omega \right)=2\int_{0}^{\infty }G\left(\tau \right)\cos\left(\omega \tau \right)d\tau$$
Substituting Equation (11) into Equation (12), and after the integral operation, the negative frequency portion is discarded. The power spectrum function of the coherent mixing signal is [16]
(13)$$\begin{array}{ll} G\left(\omega \right)= & 4A^{2}\delta \left(\omega \right)+\frac{1}{2}\beta^{2}e^{-\Delta w\tau_{d}}\delta \left(\omega -\delta_{1}\right)+\frac{1}{2}\beta^{2}e^{-\Delta w\tau_{d}}\delta \left(\omega -\delta_{2}\right)\\ & +\frac{\beta^{2}\Delta w}{2{\left(\omega -\delta_{1}\right)}^{2}+2{\left(\Delta w\right)}^{2}}+\frac{\beta^{2}\Delta w}{2{\left(\omega -\delta_{2}\right)}^{2}+2{\left(\Delta w\right)}^{2}}\\ & -\frac{\beta^{2}\Delta w\;e^{-\Delta w\tau_{d}}}{2{\left(\omega -\delta_{1}\right)}^{2}+2{\left(\Delta w\right)}^{2}}\left[\frac{\sin\left(\omega -\delta_{1}\right)\tau_{d}}{\omega -\delta_{1}}+\cos\left(\omega -\delta_{1}\right)\tau_{d}\right]\\ & -\frac{\beta^{2}\Delta w\;e^{-\Delta w\tau_{d}}}{2{\left(\omega -\delta_{2}\right)}^{2}+2{\left(\Delta w\right)}^{2}}\left[\frac{\sin\left(\omega -\delta_{2}\right)\tau_{d}}{\omega -\delta_{2}}+\cos\left(\omega -\delta_{2}\right)\tau_{d}\right] \end{array}$$
From the above equation, the power spectrum of the dual-frequency Doppler shift signal is related to the laser line width (Δw), delay time ($\tau_{d}$), Doppler shift (δ), and correlation time. When $\tau_{d}\gg \tau_{c}$, Equation (13) can be simplified to
(14)$$G\left(\omega \right)=\frac{\beta^{2}\Delta w}{2{\left(\omega -\delta_{1}\right)}^{2}+2{\left(\Delta w\right)}^{2}}+\frac{\beta^{2}\Delta w}{2{\left(\omega -\delta_{2}\right)}^{2}+2{\left(\Delta w\right)}^{2}}$$
Theoretically, the dual-frequency coherent signal has two power spectrum peaks at $\delta_{1}$ and $\delta_{2}$, but when the Doppler shift is small, the resolution of the power spectrum peak becomes difficult due to the line width and noise.

## 4. Numerical Analysis

In this section detailing the parameters in the numerical analysis, the wavelength of the laser is λ = 532 nm, the amplitude ratio is β=0.1, and the frequency difference of the frequency shifter is $f_{IF}\;=\;2\;\mathrm{GHz}$, 3 GHz and 4 GHz. The influence of the line width and motion speed on the dual-frequency optical power spectrum is analyzed according to the relationship between the delay time ($\tau_{d}$) and the coherence time ($\tau_{c}$).

### 4.1. Delay Time is Less than Coherence Time $\tau_{d}<\tau_{c}$

Firstly, the influence of the line width of the light source on the Doppler frequency difference resolution is analyzed. Assume that the target motion velocity is *V* = 100 m/s, the delay time is $\tau_{d}=0.1\tau_{c}$, and the frequency shift is $f_{IF}=2\;\mathrm{GHz}$, 3 GHz and 4 GHz. The numerical calculation results obtained using Equation (13) are shown in the Table 1 and Figure 2. In the numerical simulation process, the maximum frequency of the dual-frequency power spectrum and the frequency corresponding to the maximum point are first obtained, and then, the absolute value is taken using the difference corresponding to the frequency of the maximum point. Finally, the calculated Doppler frequency difference is obtained.

From the numerical calculation results, the theoretical Doppler frequency difference is 1.333 kHz when the frequency shift is $f_{IF}=2\;\mathrm{GHz}$, and the Doppler frequency difference can be resolved when the line width is 7 kHz, which cannot be resolved when the line width is greater than 7 kHz. When the frequency shift is $f_{IF}=3\;\mathrm{GHz}$, the theoretical Doppler frequency difference is 2 kHz, and the Doppler frequency difference can be distinguished when the line width is 10 kHz, but cannot be distinguished when the line width is greater than 10 kHz. When the frequency shift is $f_{IF}=4\;\mathrm{GHz}$, the theoretical Doppler frequency difference is 2.667 kHz, and the Doppler frequency difference can be distinguished when the line width is 14 kHz. Although, it cannot be distinguished when the line width is greater than 14 kHz. It can be further noted, that when the theoretical Doppler frequency difference is greater than 1/5 times the line width of the light source, the Doppler frequency difference can be distinguished. Although the error of the Doppler frequency difference at the critical value is large, this is a necessary condition for distinguishing the Doppler frequency difference.

Figure 2 is a power spectrum curve of line widths of different light sources when the frequency shift amount $f_{IF}=2\;\mathrm{GHz}$. It can be seen from the figure that when the line width of the light source is 3 kHz and 4 kHz, the two peaks of the power spectrum can be clearly seen. As the line width increases, the two peaks of the power spectrum gradually disappear and become a single peak. At the same time, the power spectrum peak broadens.

The line width of the light source is 1 kHz, the delay time is $\tau_{d}=0.1\tau_{c}$, and the frequency shift is $f_{IF}=2\;\mathrm{GHz}$, 3 GHz and 4 GHz. The Doppler frequency difference calculation results at different motion speeds are shown in Table 2. It can be obtained from the numerical calculation results that when the frequency shift is $f_{IF}=2\;\mathrm{GHz}$, the motion speed is 14 m/s, and the theoretical Doppler frequency difference is 186.6 Hz; the Doppler frequency difference can be distinguished. When the frequency shift is $f_{IF}=3\;\mathrm{GHz}$, the motion speed is 10 m/s, and the theoretical Doppler frequency difference is 200 Hz, then, too, the Doppler frequency difference can be distinguished. When the frequency shift is $f_{IF}=4\;\mathrm{GHz}$, the moving speed is 7 m/s, and the theoretical Doppler frequency difference is 186.7 Hz; the Doppler frequency difference can be distinguished. Furthermore, the Doppler frequency difference can be distinguished on the condition that the theoretical Doppler frequency difference is greater than 1/5 times the line width of the light source.

When the frequency shift is $f_{IF}=2\;\mathrm{GHz}$, the power spectrum at different motion speeds is shown in Figure 3. It can be seen from the figure that when the moving speed is 10 m/s, 11 m/s and 12 m/s, the power spectrum appears as a single peak. As the speed increases, the power spectrum peak decreases, and the two peaks of the power spectrum become increasingly evident.

The power spectrum of the dual-frequency coherent mixing signal is shown in Figure 4. The target motion speed is 100 m/s, the source line width is 3 kHz, and the delay times are $\tau_{d}=0.1\tau_{c}$, $\tau_{d}=0.3\tau_{c}$, $\tau_{d}=0.5\tau_{c}$, $\tau_{d}=0.7\tau_{c}$. It can be seen from the figure that when the delay time is less than the coherence time of the light source, the power spectrum of the dual-frequency coherent mixing signal is broadened as the delay time increases; however, the resolution of the dual-frequency power spectrum is small.

### 4.2. Delay Time is Larger than Coherence Time $\tau_{d}>>\tau_{c}$

When the delay time is greater than the source coherence time, the power spectrum of the dual Doppler shift signal can be approximated using Equation (14). It can be seen from the formula, the signal power spectrum is independent of the delay time $\tau_{d}$, and is only related to the line width of the light source and the Doppler shift.

The line widths of the light source are 1 kHz, 3 kHz, 5 kHz and 7 kHz, and the target motion speed is 50 m/s. The Doppler frequency difference Δδ=666 Hz. The numerical simulation results are shown in Figure 5. As can be seen from the figure, the power spectrum curve of the dual-frequency coherent mixing signal is close to the Lorentz type. As the line width of the light source increases, the power spectrum broadens, the peak decreases, and the peak resolution of the dual-frequency Doppler power spectrum decreases. When the line width of the source is about 5 times the Doppler frequency difference, the peak of the dual-frequency power spectrum is close to the resolution limit.

In summary, under the condition that the Doppler frequency difference is small, the power spectrum resolution of the dual-frequency coherent mixing signal is closely related to the line width of the light source, and has nothing to do with the delay time. When the line width of the light source is about five times the Doppler frequency difference, the peak of the power spectrum is close to the resolution limit. If the Doppler frequency difference is less than the power spectrum resolution limit, the microwave beat signal cannot be obtained by optical second-order mixing or electric signal mixing. Therefore, in practical applications, the frequency difference of the dual-frequency light should be as large as possible, so that the Doppler frequency difference is large, and the power spectrum resolution limit is avoided.

## 5. Conclusions

This paper mainly analyzes the power spectrum resolution of dual-frequency coherent mixing signals when the Doppler frequency difference is small. The power spectrum function formula of the dual-frequency coherent mixing signal is obtained using statistical theory. The power spectrum of the signal is related to the line width of the source, the delay time and the Doppler frequency difference. When the Doppler frequency difference is small, the power spectrum of the dual-frequency signal has a resolution limit. If it is less than the power spectrum resolution limit, the microwave beat signal cannot be obtained by optical second-order mixing or electric signal mixing. Increasing the frequency shift is easier to reduce the line width of the light source, and increasing the frequency shift can effectively improve the resolution of the Doppler frequency difference. Therefore, the large frequency difference dual-frequency light source should be used in practical applications, so that the Doppler frequency difference is greater than the power spectrum resolution limit.

The analysis results in this paper provide a theoretical reference for laser line width selection, frequency shift selection and velocity measurement limits, for use in the application of dual-frequency laser coherent detection technology.

## Figures and Tables

**Figure 1 sensors-19-05264-f001:**
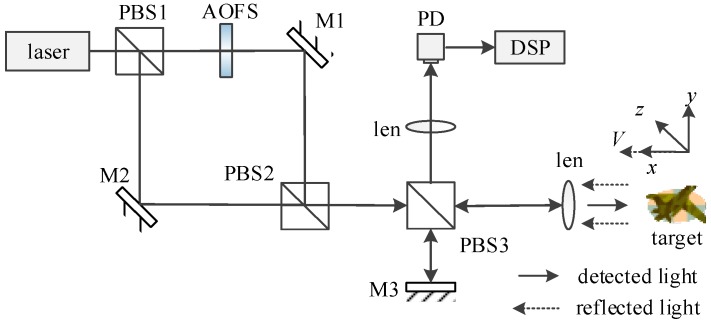
Structure diagram of moving target detection system based on four-light coherence mixing technology.

**Figure 2 sensors-19-05264-f002:**
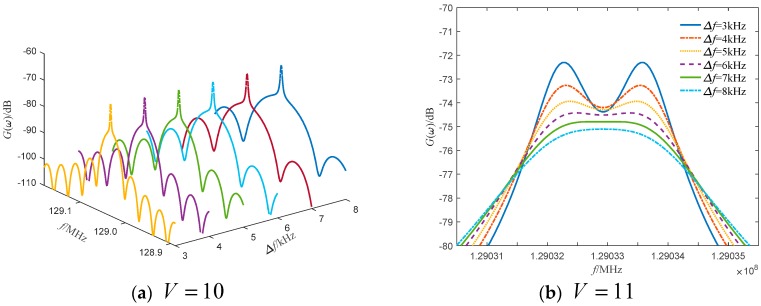
Power spectrum curves of different line widths of $\tau_{d}=0.1\tau_{c}$.

**Figure 3 sensors-19-05264-f003:**
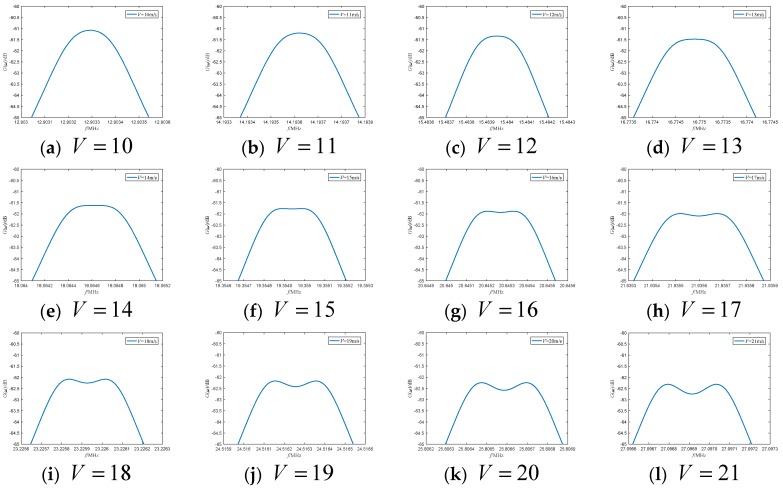
The power spectrum curve at different motion speeds when the frequency shift is 2 GHz.

**Figure 4 sensors-19-05264-f004:**
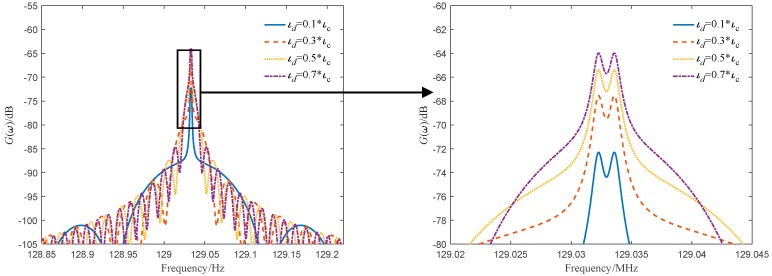
The power spectrum curves of different delay time when the line width of the light source is 3 kHz.

**Figure 5 sensors-19-05264-f005:**
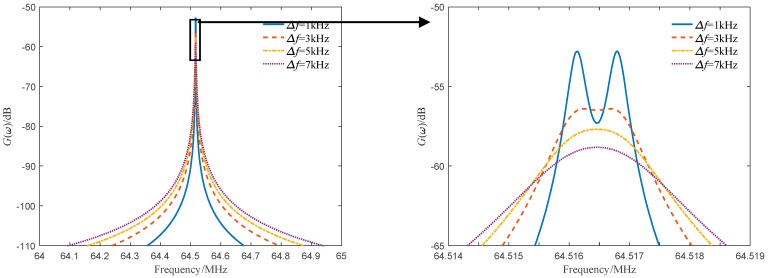
Power spectrum curves of different light source line widths of *V* = 50 m/s.

**Table 1 sensors-19-05264-t001:** The Doppler frequency difference calculation results of line widths of different light sources when the frequency shift is 2 GHz, 3 GHz and 4 GHz, respectively.

Line Width/kHz	2 GHz		3 GHz		4 GHz	
	Theoretical Doppler Frequency Difference/kHz	Calculated Doppler Frequency Difference/kHz	Theoretical Doppler Frequency Difference/kHz	Calculated Doppler Frequency Difference/kHz	Theoretical Doppler Frequency Difference/kHz	Calculated Doppler Frequency Difference/kHz
1	1.333	1.333	2	2	2.667	2.667
2	1.333	1.325	2	1.998	2.667	2.665
3	1.333	1.298	2	1.988	2.667	2.661
4	1.333	1.232	2	1.966	2.667	2.651
5	1.333	1.107	2	1.922	2.667	2.631
6	1.333	0.887	2	1.848	2.667	2.595
7	1.333	0.426	2	1.736	2.667	2.541
8	1.333	\	2	1.570	2.667	2.463
9	1.333	\	2	1.330	2.667	2.357
10	1.333	\	2	0.952	2.667	2.215
11	1.333	\	2	\	2.667	2.025
14	1.333	\	2	\	2.667	0.853
15	1.333	\	2	\	2.667	\

**Table 2 sensors-19-05264-t002:** The Doppler frequency difference calculation results of different motion speeds when the frequency shift is 2 GHz, 3 GHz and 4 GHz.

Speed m/s	2 GHz		3 GHz		4 GHz	
	Theoretical Doppler Frequency Difference/kHz	Calculated Doppler Frequency Difference/kHz	Theoretical Doppler Frequency Difference/kHz	Calculated Doppler Frequency Difference/kHz	Theoretical Doppler Frequency Difference/kHz	Calculated Doppler Frequency Difference/kHz
6	0.080	\	0.120	\	0.160	\
7	0.0933	\	0.140	\	0.1867	0.040
8	0.1067	\	0.160	\	0.2133	0.129
9	0.120	\	0.180	\	0.240	0.180
10	0.1334	\	0.200	0.090	0.2667	0.220
11	0.1466	\	0.220	0.140	0.2933	0.250
12	0.160	\	0.240	0.180	0.320	0.290
13	0.1734	\	0.260	0.211	0.3467	0.324
14	0.1866	0.089	0.280	0.240	0.3733	0.354
15	0.200	0.095	0.300	0.267	0.400	0.384
16	0.2134	0.129	0.320	0.292	0.4267	0.413
17	0.226	0.156	0.340	0.316	0.4533	0.442
18	0.2240	0.180	0.360	0.339	0.480	0.471
19	0.2533	0.201	0.380	0.362	0.5067	0.498
20	0.2666	0.222	0.400	0.384	0.5333	0.526

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
