# Peer review of "The Effect of Light Source Line Width on the Spectrum Resolution of Dual-Frequency Coherent Detection Signals"

_sensors, 2019, doi:10.3390/s19235264_

Round 1
Reviewer 1 Report
The authors describe the effect of laser source line width on dual-frequency coherent detection systems, and in particular the interplay between coherence time (line width), time delay (range) and target speed resolution (Doppler-shift difference).
In the first part (sections 2 and 3), the predicted power spectrum of the detected signal for arbitrary target distance, target speed, and laser coherence time is computed. There are various omissions and mistakes in the equations that make it difficult to follow and confirm the results (e.g. what is η in Eq. (1); what are Es1 and Es2 in Eq. (2); there is something missing in Eq. (3); is Eq.(9) valid for all τ?)
Provided Eq. (12) is correct, the conclusion is then that the broader the line width of the laser source, the poorer the resolution of the Doppler-shift difference measurement. It would be interesting to quantify this statement: for example, what is the slowest speed that can be measured as a function of laser line width and intermediate frequency? What is the interplay between speed resolution, laser line width, range and intermediate frequency (ωIF)? etc.
Author Response
Point 1: In the first part (sections 2 and 3), the predicted power spectrum of the detected signal for arbitrary target distance, target speed, and laser coherence time is computed. There are various omissions and mistakes in the equations that make it difficult to follow and confirm the results (e.g. what is η in Eq. (1); what are Es1 and Es2 in Eq. (2); there is something missing in Eq. (3); is Eq.(9) valid for all τ?)
Response 1: I have made serious changes and improvements to the errors and omissions in the formula you pointed out. E.g. in the formula (1), b represents the amplitude ratio of the light field E01 to the light field E02, and is the diffraction efficiency of the frequency shifter. and are reflected light field signals in Eq.(2). is the target speed of movement. is the speed of light in Eq.(3). The in the formula (9) is the time difference of the light field in the autocorrelation function, so Eq.(9) is valid for all
Point 2: Provided Eq. (12) is correct, the conclusion is then that the broader the line width of the laser source, the poorer the resolution of the Doppler-shift difference measurement. It would be interesting to quantify this statement: for example, what is the slowest speed that can be measured as a function of laser line width and intermediate frequency? What is the interplay between speed resolution, laser line width, range and intermediate frequency (ωIF)? etc.
Response 2: According to your suggestion, this paper focuses on the Doppler frequency difference of linewidths of different light sources when the frequency shift is 2GHz, 3GHz and 4GHz respectively, and obtains the corresponding quantitative results.At the same time, the Doppler frequency difference resolution under different motion speeds is analyzed and the quantitative results are given. From the analysis results, it can be concluded that the minimum theoretical Doppler frequency difference that can distinguish the Doppler frequency difference is 1/5 times the line width of the light source.
The minimum velocity resolution of the Doppler frequency difference at 2 GHz is 14 m/s. The minimum speed resolution of the Doppler frequency difference at 3 GHz is 10 m/s. The minimum speed resolution of the Doppler frequency difference at 4 GHz is 7 m/s.

Reviewer 2 Report
The paper presents a theoretical analysis of dual-frequency laser coherent detection by calculating the resolution of the power spectrum. In particular, the authors pay attention to small Doppler shifts (slowly moving objects) and analyze the influence of the delay time, as well as linewidth and coherence time of the laser source. The main findings are that the delay time has practically no influence, as it only affects the peak of the power spectrum. In contrast, increasing source line width results in a broadened spectrum and decreased peak value. The resolution limit of the power spectrum is reached if the source linewidth increases to about five times the Doppler frequency difference, implying that a large frequency difference of the dual-frequency light source (large AOM modulation frequency) should be used.
While these conclusions are important for the design of such a setup, I have to question both the novelty and significance of this study. The main result of the derivation in chapters 2 and 3, namely Equ. 12, is essentially the same as Equ. 18 in Ref. 13 (Acta Phys. Sin. 2016, 65(8): 84206). The fact that the derivation in Ref. 13 is for a single frequency, while Equ. 18 in the presented paper builds the sum for two source frequencies is not significant. The following numerical analysis in chapter 4 provides valuable insights, but it seems that this can easily be performed based on Ref. 13. Unless there are special tricks required, which then should be mentioned in the paper, I do not see any novelty of this paper. If the novelty-threshold for MDPI Sensors is not reached, I cannot suggest publication in this journal. In any case, it must at least be mentioned that the derivation is in close analogy to Ref 13 (or any other paper with this derivation, preferable in English)!
In addition, I have the following concerns:
1) The derivation cannot be followed because the references in key positions are to papers in Chinese language:
Line 77: “…the autocorrelation function of the stochastic stationary signal and the corresponding power spectral density function are Fourier transform pairs [13].”
line 88: “According to the Wiener-Sinchin theorem [13], the power spectrum of the autocorrelation function is…”
2) There are many obvious mistakes and undefined quantities in the derivation:
Equ 1: \eta is not defined
Equ 2: “b” and “(1-b) \eta” seem to be missing
Equ 3: \omega is not defined, shouldn’t that be \omega_1 and \omega_2, respectively?
Equ 3: The velocity “V” is not defined, is this parallel or perpendicular to the laser beam direction?
Equ 6: \eta^2 should be \eta
Equ 7: The step from Equ 4/5 to Equ 7 is not clear and requires a reference or explanation.
Equ7: a plus symbol is missing between the third and fourth term
Equ 7: A is not defined
Equ 7: The step from Equ 7 to Equ 10 is not clear and requires a reference or explanation.
Line 90: “Put Equation (13) into Equation (14)” should read “Put Equation (10) into Equation (11)”
3) Many sentences or phrases are not clear or incorrect:
Line 28: “...and high Doppler of laser heterodyne detection system …”
Line 28: “The contradiction between frequency shifts.” (no verb)
Line 41: “The effect of speed on the resolution of dual-frequency Doppler signals and numerical simulation analysis”
Line 89: “...Wiener-Sinchin theorem …” to “...Wiener-Khinchin theorem …”
Line 93: “...and correlation …” (correlation time?)
Line 118: “…when the line width of the line widths is 3kHz” -> “…when the line width of the LIGHT SOURCE is 3kHz”
Line 125: “...delay time \tau_d = 0.1 \tau_c …” This is in the \tau_d >> \tau_c limit, why is \tau_d = 0.1 \tau_c assumed here?
Line 131: “When the linewidth of the source is about 5 times the Doppler shift, …” should read “When the linewidth of the source is about 5 times the Doppler FREQUENCY DIFFERENCE, …”
Line 138: “If it is less than…” what does “it” refere to?
Chapter 5: The conclusion has the same content as the summary before (this is minor).
References: Check the references! (e.g., Ref 6 is missing the journal name, Ref 15 is missing the author names, …)
4) Figure 1 needs improvements:
a) Mirrors M2 and M3 (?) need to be drawn.
b) is the fundamental of the AOM blocked? If yes, this should be indicated.
c) The coordinate system is left-handed. Please discuss the directionality of V in the text!
Author Response
Response 1: Thank you very much for your valuable comments. Your sincere advice has made me realize my deficiencies and let me improve and make progress.
The Equ. 12 essentially the same as Equ. 18 in Ref. 13. However, this paper is different from the research target of reference (13). Perhaps this is related to the unclear expression of the initial manuscript and the incomplete analysis.
Doppler shift resolution of dual-frequency light is the key to dual-frequency laser coherent detection.In the case of high-speed motion and large amount of frequency shift, the Doppler frequency difference of the dual-frequency light is large, and the Doppler shift resolution of the dual-frequency light is easy to implement. However, in the actual application process, there is still insufficient motion speed or small amount of frequency shift, which causes the Doppler spectrum to be indistinguishable due to the small Doppler frequency difference. Although this is a special critical situation, it is necessary to master the correlation between Doppler shift resolution and Doppler frequency difference and source linewidth. However, there is still a lack of theoretical analysis for this aspect.
In the paper, when the frequency shift is 2 GHz, 3 GHz and 4 GHz, the Doppler frequency difference resolution of line width and motion velocity of different light sources is numerically quantized. From the results, the Doppler frequency difference can be obtained. The Pule frequency difference is 1/5 times the line width of the light source. This conclusion is an important argument in this paper.
The abstract, introduction and conclusions of this paper have been revised.
Response 2: I am sorry for the inconvenience caused by the Chinese reference.
In response to the above questions, I added corresponding English references in the corresponding positions.
After edited:
Line 91:” The autocorrelation function of the light field can be expressed as [15]”.
Line 104:” The power spectrum function of the coherent mixing signal is[18]”.
Response 3:
Thank you very much for your accurate and meticulous comments and suggestions.In accordance with the errors and omissions you pointed out, I made serious changes and improvements.
Equ 1: \eta is the frequency shifter diffraction efficiency.
Equ 2: The amplitude of the reflected light is much smaller than that of the local oscillator. Therefore, the amplitude difference between the two reflected lights is very small, so the amplitude difference between the two reflected lights is ignored in the text.
Equ 3:The \omega in Equation 3 has been changed to \omega_1 and \omega_2.
V is the target speed of movement.When the target moves toward the detection system, the velocity V is positive; when moving away from the detection system, the velocity V is negative. In this paper, it is assumed that the target is moving forward in the x-axis, that is, the Doppler shift is positive.
Equ 6: \eta^2 changed to \eta
Equ 7: Added English reference [15] from formula (4)(5) to formula (7)
Redefining A,
Added reference [13][15] from formula (7) to formula (10)
line 90(line 105, after modification): “Put Equation (11) into Equation (12),”
Response 4:
Line 28:”The dual-frequency laser coherent detection technology can reduce the high Doppler shift of high-speed moving targets to microwave frequency by microwave beat method [5,6], which solves the contradiction between the low response frequency and larger Doppler frequency shifts.”
Line 41: The introduction has been revised.
Line 89(line 104): “...Wiener-Sinchin theorem …”changed to “...Wiener-Khinchin theorem …”
Line 93(line 108): “...and correlation …” changed to “correlation time”
Line 118(line 170): “…when the line width of the line widths is 3kHz”changed to “…when the line width of the LIGHT SOURCE is 3kHz”
Line 125(line 177):deleted “...delay time \tau_d = 0.1 \tau_c …”
Line 131(line 182): “When the linewidth of the source is about 5 times the Doppler shift, …” changed to “When the linewidth of the source is about 5 times the Dopplerfrequency difference, …”
Line 138(189): edited “ If the Doppler frequency difference is less than the power spectrum resolution limit, the microwave beat signal cannot be obtained by optical second-order mixing or electric signal mixing.”
Chapter 5: “ This paper mainly analyzes the power spectrum resolution of dual-frequency coherent mixing signals when the Doppler frequency difference is small. The power spectrum function formula of the dual-frequency coherent mixing signal is obtained by statistical theory. The power spectrum of the signal is related to the linewidth of the source, delay time and Doppler frequency difference.When the Doppler frequency difference is small, the power spectrum of the dual-frequency signal has a resolution limit. If it is less than the power spectrum resolution limit, the microwave beat signal cannot be obtained by optical second-order mixing or electric signal mixing. Increasing the frequency shift is easier to reduce the line width of the light source, and increasing the frequency shift can effectively improve the resolution of the Doppler frequency difference.Therefore, the large frequency difference dual-frequency light source should be used in practical applications, so that the Doppler frequency difference is greater than the power spectrum resolution limit.
The analysis results in this paper provide a theoretical reference for laser linewidth selection, frequency shift selection and velocity measurement limits in the application of dual-frequency laser coherent detection technology”.
References: Improved reference information and added references [16]
Response 4:
Redrawn Figure 1
Redraw the mirrors M1, M2 and M3 The AOM in the figure should be modified for AOFS(Acousto-optic frequency shifter), Assume that the frequency shift of the AOFS(acousto-optic frequency shifter) is positive. The left-handed coordinate system in the figure is changed to the right-handed coordinate system.

Round 2
Reviewer 1 Report
This is a significant improvement with respect to the previous version. Some minor language issues remain, and perhaps the results in the large tables can be presented in more compact form. I think also that an explanation should be given about what "Calculated Doppler frequency difference" means in the tables (i.e. how do you get to the values in that column).
Author Response
Point 1:Some minor language issues remain, and perhaps the results in the large tables can be presented in more compact form. I think also that an explanation should be given about what "Calculated Doppler frequency difference" means in the tables (i.e. how do you get to the values in that column).
Response 1:
According to your suggestion, I have modified seven language expressions by checking this article, such as line 11, line 21, line 38, line 41, line 44, line 80, line 121.
The data in Tables 1 and 2 is simplified.In Table 1, the original 500 Hz line width interval was replaced with a 1 kHz line width interval.The target range of motion speeds in Table 2 retains representative 6-20 m/s data.
Line 121:The method of obtaining the "calculated Doppler frequency difference" is explained.“In the numerical simulation process, the maximum frequency of the dual-frequency power spectrum and the frequency corresponding to the maximum point are first obtained, and then the absolute value is taken by the difference corresponding to the frequency of the maximum point, and finally the calculated Doppler frequency difference is obtained.”

Reviewer 2 Report
The manuscript has significantly improved and should be published.
Author Response
The data in Tables 1 and 2 is simplified.In Table 1, the original 500 Hz line width interval was replaced with a 1 kHz line width interval.The target range of motion speeds in Table 2 retains representative 6-20 m/s data.
Line121:
The method of obtaining the "calculated Doppler frequency difference" is explained.
“ In the numerical simulation process, the maximum frequency of the dual-frequency power spectrum and the frequency corresponding to the maximum point are first obtained, and then the absolute value is taken by the difference corresponding to the frequency of the maximum point, and finally the calculated Doppler frequency difference is obtained.”
